# In-Season Weightlifting Training Exercise in Healthy Male Handball Players: Effects on Body Composition, Muscle Volume, Maximal Strength, and Ball-Throwing Velocity

**DOI:** 10.3390/ijerph16224520

**Published:** 2019-11-15

**Authors:** Souhail Hermassi, Mohamed Souhaiel Chelly, Nicola Luigi Bragazzi, Roy J Shephard, René Schwesig

**Affiliations:** 1Sport Science Program, College of Arts and Sciences, Qatar University, Doha 2713, Qatar; 2Research Unit (UR17JS01) Sport Performance, Health & Society, Higher Institute of Sport and Physical Education, Ksar-Saîd, University of “La Manouba”, Tunis 2010, Tunisia; csouhaiel@yahoo.fr; 3Laboratory for Industrial and Applied Mathematics (LIAM), Department of Mathematics and Statistics, York University, 4700 Keele Street, Toronto, ON M3J 1P3, Canada; robertobragazzi@gmail.com; 4Faculty of Kinesiology and Physical Education, University of Toronto, Toronto, ON M5S 1A1, Canada; royjshep@shaw.ca; 5Department of Orthopaedic and Trauma Surgery, Martin-Luther-University Halle-Wittenberg, 06097 Halle, Germany; rene.schwesig@uk-halle.de

**Keywords:** muscle volume, weightlifting exercises, maximal strength, throwing velocity, healthy handball players

## Abstract

This study assessed the impact of 8 weeks biweekly in-season weightlifting training on the strength, throwing ability, and body composition of healthy male handball players. Twenty players (age: 21.2 ± 0.7 years, height: 1.83 ± 0.08 m, body mass: 83.3 ± 7.5 kg, body fat: 13.2 ± 1.4%, upper limb muscle volume: 3.16 ± 0.16 L) were randomly allocated between experimental (EG) and control (CG) groups. Measures of one-repetition maximal strength included bench press, pull-over, snatch, and clean and jerk. Throwing velocity was investigated by standing, running, and jump throws, and the power of the upper limbs was estimated from the total distance of a 3-kg medicine ball overhead throw. Muscle volumes were estimated anthropometrically. Training sessions comprised 3–4 sets of explosive weightlifting exercise at 75%–90% of 1RM (repetition maximum). Significant interaction effects (time x group) were found for all strength and throwing variables, ranging from η_p_^2^ = 0.595 (pull-over) to η_p_^2^ = 0.887 (medicine ball throw), with the largest between-group difference (more than 40%, Δd = 6.65) and effect size (d = 6.44) for the medicine ball throw, and the smallest (about 23%, Δd = 1.61) for the standing shot performance. Significant interaction effects were also detected for all anthropometric parameters (body mass: η_p_^2^ = 0.433; body fat: η_p_^2^ = 0.391; upper limb muscle volume: η_p_^2^ = 0.920, with an almost 20% gain of muscle volume). It can be concluded that 8 weeks of biweekly in-season weightlifting training yielded substantial increases of muscle volume, maximal strength of the upper limbs, and ball throwing velocity in healthy handball players relative to their standard training program.

## 1. Introduction

Handball is a highly demanding intermittent sport, involving multiple high-intensity short runs [1,2,3], frequent body contacts [1,2,3], and other high-intensity explosive strength and power actions. Creativity in combination with speed-jumping, turning, changing pace, and ball throwing makes this sport very attractive but tough to play [1,2,3,4].

Specific studies of this issue have examined the influence of speed-strength programs on throwing ball velocity [5], or the relationship between throwing velocity and isokinetic strength [6], rather than the impact on jumping ability, sprint performance, and maximal dynamic strength [1,2,4,7]. Handball coaches and scientists agree that the main determinants of throwing velocity are technique, the timing of movement in consecutive body segments, and the strength and power of the upper and lower limbs [1,2,3,4,5]. Each of these factors can be improved by appropriate training, particularly resistance programs [1,2,7,8]. Chelly et al. [5] previously highlighted the contribution of the lower limbs to throwing ability, underlining that coaches should include strength and power programs not only for the shoulders and arms, but also for the lower limbs. Biweekly training of this type seems sufficient to induce substantial gains not only in peak power output and dynamic strength, but also in handball throwing velocity [7,8].

In-season strength and conditioning have thus been advocated to preserve and increase the strength and power of players [1,2,4,7,9]. Both strength and power can be enhanced by resistance training [3,4]. Some studies have found associated improvements in explosive actions [2,3,4], but others have seen no significant increases [4], suggesting that strength training may lack elements important to explosive movements, including eccentric overloading, segmental coordination, and specificity of joint angles and angular velocities [3].

However, there is a dearth of information concerning the most effective way to optimize physical performance in handball, with little known about possible interference between the different potential components of a training regimen [4,9]. Of crucial interest are modalities that are simple and effective, yet less expensive and easier to use than weight machines or free weights. One such option is weightlifting, now widely adopted to enhance both neuromuscular power and performance in team sports [3,4]. Such training involves ballistic resistance exercises; gains in strength are sometimes smaller than with heavy resistance programs, but at the same time players may develop greater increases in powerful movements as analyzed and interpreted by the force–time curve for rapid force production [3,10,11].

Weightlifting movements and their derivatives are now widely adopted to enhance neuromuscular power and explosive performance in team sports [12,13]. Such training involves ballistic resistance exercises that often yield smaller increases in maximal strength than heavy resistance programs, but at the same time it develops greater increases in powerful movements as analyzed and interpreted by the force–time curve for rapid force production [10,14]. Clean-and-jerks and snatches are examples of this approach to the development of muscular power in team sports. In order to optimize performance, an athlete should increase strength as well as the rate at which force can be developed at the hip, knee, and ankle joints. High-velocity resistance exercises are particularly useful in developing more powerful movements at the speeds required during actual play [15,16].

The present investigation assessed improvements in the muscular strength and ball-throwing velocity of healthy male handball players resulting from in-season weightlifting training. The hypothesis tested was that male handball players who replaced a part of their standard training with an 8-week program of biweekly weightlifting would increase the muscular strength and power of their limbs and their ball-throwing velocity relative to players who maintained their normal in-season regimen.

## 2. Materials and Methods

### 2.1. Ethical Clearance

All procedures were approved by the committee for the ethical use of human subjects of the Research Unit Sport Performance, Health and Society at the University of La Manouba, Tunisia, according to current national and international laws and regulations. Participants were thoroughly advised about the experimental design of the present study and its potential risks and benefits, and they then signed their written informed consent. They were free to withdraw from the investigation without penalty at any time.

### 2.2. Procedures and Evaluations

Participants were healthy handball players who were in a good health with no evidence of past or present metabolic disorders, diabetes mellitus, or cardiovascular abnormalities. We excluded individuals undergoing any type of medical treatment and those who were obese (body mass index > 30 kg/m^2^). A questionnaire regarding medical history, age, height, body mass, training characteristics, injuries, playing position, performance experience, and level of competition was first administered. The team physician then carried out a medical examination, focusing on orthopedic conditions and clinical disorders that might preclude participation in resistance exercises. Twenty healthy male handball players from the Second National League (age: 21.2 ± 0.7 years, height: 1.83 ± 0.08 m, body mass: 83.3 ± 7.5 kg, body fat: 13.2 ± 1.4%, mean playing experience 10.1 ± 0.5 years) were randomly allocated between an experimental (EG) and a control (CG) group using a random number generator.

All participants engaged in the same training sessions, carefully supervised by the teams’ coaches, from the beginning of the competitive season (September) until the end of the trial (March). During the playing season, the normal routine consisted of six 90-min training sessions per week, plus a competitive game played at the weekend. Physical conditioning two times per week was aimed at strength development; it incorporated high-intensity interval training and gymnastics. Anaerobic training consisted of plyometric and sprint training drills, and aerobic fitness was developed using small-sided games. Training sessions consisted mainly of technical–tactical skill development (60% of the session time) and strength and conditioning routines (40% of the session time).

During the 8-week intervention, the CG maintained this standard pattern of training, but twice per week the EG replaced their technical–tactical skill development by a dynamic weightlifting program designed to maximize increases in strength. They wore handball shoes to adapt and fit their specific efforts to handball activity. Training focused upon the extensor muscles of the upper limbs (bench-press, pull-over, snatch, and clean-and-jerk exercises at 60%–85% of the individual’s personal one repetition maximum [1RM]). All of these exercises required a series of successive eccentric–concentric loaded contractions, conducted at a slow velocity; 3–8 repetitions were carried out per set, and 3–6 sets were interspersed with rest intervals lasting 3 to 4 min.

Test measurements were conducted on a regular indoor handball court, at the same time of day (from 5:00 p.m. to 7:00 p.m.) and under the same experimental and environmental conditions (temperature 20.5 ± 0.5 °C, relative humidity 60 ± 5%), at least 3 days after a competition. Participants maintained their normal intake of food and fluids, with the exception that they abstained from physical activity, drank no caffeine-containing beverages, and ate no food for 1 day, 4 h, and for 2 h before testing, respectively.

### 2.3. Testing Schedule

After a general warm-up, a specific warm-up was performed for each test procedure. All maximal efforts were preceded by 5 min of low intensity running, 3 × 30 m progressive accelerations, and a maximal 30 m sprint, interspersed with 3 min-periods of passive recovery. Trunk rotation, trunk side-bends, trunk wood-chops, internal and external rotary movements of the shoulder, push-ups and 8 to 10 free-ball throws were performed prior to the throwing tests.

Three similar sets of tests were integrated into the weekly training schedules. The first set of tests, completed two weeks before the intervention, familiarized participants with procedures, and allowed an assessment of the 2-week test-retest reliability of measurements. The second and third sets were completed in the week prior to and immediately following the 8-week intervention. Each set of tests was administered on three non-consecutive days, with technicians blinded to group allocation.

On the first definitive test day, anthropometric measurements were completed, followed by the 1RM bench press (1RM_BP_) and 1RM snatch (1RM_snatch_) tests. On the second day, 1RM clean and jerk (1RM_cj_) tests were assessed. On the third day, handball throwing and the 1RM pull-over (1RM_PO_) were evaluated (Figure 1).

### 2.4. Experimental Approach to the Problem

The aim of the study was to examine whether 8 weeks of biweekly in-season weightlifting training would improve the muscular strength and power of the upper limbs and the ball-throwing velocity compared to the standard in-season training regimen. A sample of 20 male handball players was randomly assigned to either an experimental weightlifting group (EG; *n* = 10) or a control group who maintained their standard in-season training regimen (CG; *n* = 10). All participants had already been training for five months, and were four months into their competitive season. Two familiarization trials were completed in the two weeks before the intervention. Data were obtained both before modification of training, and after completion of the trial. Three to 6 attempts were needed to reach the 1RM for *Bench Press, Bench snatch, Clean-and-Jerk, and Pull-over.*

### 2.5. Anthropometry

The muscle volume of the upper and lower limbs was estimated from circumferences and skin-fold thickness at different levels of the thigh and the calf, the arm and the forearm, as well as the length of the upper and lower limbs and the breadth of the humeral and femoral condyles, using validated procedures [17,18]. The percentage of body fat was estimated from measurements of biceps, triceps, and subscapular and suprailiac skinfolds [19].

### 2.6. 1RM Bench Press (1RM_BP_)

The bench press involved muscles highly related to overhand throwing, including the triceps brachialis and the pectorals. Participants performed a warm-up protocol of 5-min cycling with a 60-W load, stretching of the lower limb muscles and 2 min of jumping exercises (skipping (6 m), double limb ankle hops (6 reps), standing jump and reach (5 reps), and split squat jump (5 reps)). The stretching exercises—involving gastrocnemius, quadriceps, hip flexors, hamstrings, and gluteal muscles—were performed twice, holding each stretch for about 15 s and alternating legs to recover adequately.

A Smith machine was utilized, with the barbell attached at both ends. Linear bearings allowed only vertical movements. The 1RM_BP_ was performed using a theoretical maximal load determined during test familiarization. The bar was positioned 0.3 m above the chest and supported by the bottom stops of the measuring device [7]. A series of eccentric and concentric actions were performed from the starting position, maintaining the shoulders abducted at 90 º and allowing no bouncing or arching of the back. Warm-up comprised five repetitions at 40%–60% of the perceived maximum. From four to five attempts with 2-min rest intervals were conducted, until the participant was unable to extend his arms fully. The barbell was then loaded to 90% of the pretest 1RM. Two consecutive tests were carried out and, if 2 repetitions were completed, a load of 5 kg was added after a recovery interval of 3 min. When the participant had performed 2 successful repetitions of his pretest RM value, further loads of 1 kg were added after the recovery interval. If the second repetition could not be completed with the new loading, the corresponding load was considered as the individual’s 1RM [7].

### 2.7. 1RM Bench Snatch (1RM_snatch_)

The power snatch aims to lift the loaded barbell from the ground to an overhead position in continuous motion [20], with only a slight bending of the knees and hips. After a 10-min standardized warm-up, participants began with a number of warm-up lifts at 40%–60% of 1 RM. Definitive lifts were performed as follows: 2 at 70%, 2 at 80%, 1 at 90%, and 1 at 95% of the one repetition maximum (1RM) with 2–5 min rested between sets [20]. The barbell was then loaded to 90% of the pretest 1RM. Two consecutive tests were carried out, and if 2 repetitions were completed, a load of 5 kg was added after a recovery interval of 3 min. When the participant had performed 1 successful repetition of his pretest 1 RM value, further loads of 1 kg were added after the recovery interval. If the second repetition could not be completed with the new loading, the corresponding load was considered as the individual’s 1RM [20].

### 2.8. 1RM Clean and Jerk (1RM_CJ_)

Participants performed a warm-up, similar to that conducted for the 1RM power snatch, before each session. When performing the power clean and jerk, the loaded barbell was lifted with a shoulder-wide grip and the knees were initially bent. During the power clean, the barbell was moved from the floor to a racked position across the deltoids, without resting fully on the clavicles. From this shallow squat position, the participant stood up and thereafter lifted the loaded barbell onto the straightened arms to a final position with the barbell overhead [20]. During the jerk, the barbell was lifted above the head, with straight arms and legs, and the feet were kept in the same plane as the torso and barbell.

The barbell was loaded to 90% of the pretest 1RM. Two consecutive tests were carried out and, if 2 repetitions were completed, a load of 5 kg was added after a recovery interval of 3 min. When the participant had performed 2 successful repetitions of his pretest RM value, further loads of 1 kg were added after the recovery interval. If the second repetition could not be completed with the new loading, the corresponding load was considered as the individual’s 1RM [20].

### 2.9. 1RM Pull-Over (1RM_PO_)

This test is similar to the dumbbell pullover, but intensity is added to the movement by utilizing a barbell [8]. All participants had used this technique in their weekly strength training programs and were quite familiar with it. The bar was positioned 0.2 m above the participant’s chest and was supported by the bottom stops of the weight rack. Successive eccentric–concentric contractions were performed from the starting position, as described by Hermassi et al. [8]. A pretest assessment of the 1RM pull-over (1RM_PO_) was made during the final standard training session. Prior to each session, athletes performed a warm-up for 5 min of running, dynamic activities and stretching. This warm-up also included a set of five repetitions at loads of 40%–60% of their pretest 1RM. The barbell was then loaded to 90% of the pretest 1RM. Two consecutive tests were carried out and, if 2 repetitions were completed, a load of 5 kg was added after a recovery interval of 3 min. When the participant had performed 2 successful repetitions of his pretest RM value, further loads of 1 kg were added after the recovery interval. If the second repetition could not be completed with the new loading, the corresponding load was considered as the individual’s 1RM [8].

### 2.10. Handball Throwing

Handball-specific explosive strength was evaluated by a 3-step running throw and a jump throw. After a 10-min standardized warm-up, participants put resin on their hands and threw a standard handball (mass 480 g, circumference 58 cm) towards the upper right corner of the goal until three correct throws had been recorded, to a maximum of three sets of three consecutive throws. One to 2-min and 10–15 s rest intervals were allowed between sets and throws, respectively. In the jump-throw, players took a preparatory three-step run before jumping vertically and releasing the ball while in the air, behind a line 9 m from the goal. In the running throw, a preparatory run of three regular steps was made before releasing the ball, again, 9 m from the goal. Throwing time was recorded by a digital video camera (HVR to A1U DV Camcorder; Sony, Tokyo, Japan, accuracy 1 ms) positioned 2 m above and perpendicular to the ball release. Commercial software (Regavi & Regressi, Micrelec, Coulommiers, France) converted handball displacements to velocities. The reliability and validity of the software were verified [21] by using the camera (Vc) to measure the speed of rolling balls (2–14 m/s) and checking data over a 3 m distance against photoelectric cell estimates (Vpc) (GLOBUSREHAB and Sports High Tech, Articolo ERGO TIMER, Codognè, Italy). The two values were well correlated (Vc = 0.9936Vpc + 0.65; r = 0.99; *p* < 0.0001) [21], with a test–retest correlation of 1.9%. The greatest average velocity was selected for further analysis.

### 2.11. Weightlifting Training Program

Certified strength and conditioning specialists supervised each training session. A warm-up that incorporated jogging, dynamic stretching exercises, and calisthenics was conducted before each training session, and sessions ended with a 5-min cool-down that included dynamic stretching.

Subjects were encouraged to increase the amount of weight lifted gradually, until they achieved concentric fatigue. All completed a minimum of 95% of sessions. Each training session consisted of four different exercises and from 3 to 4 sets of 8–10 repetitions [11]. Exercises consisted of compound lifts involving multiarticular movements and various muscle groups, so that 4 exercises per muscle group provided sufficient stimuli. The volume and intensity of effort were selected in accordance with previous guidelines for handball players [4], and the ability of the individual [22], based on his 10-repetition-maximum for the selected resistance exercises. If the required number of repetitions could not be achieved within a set, 3 min of rest were allowed before further attempts.

The training program comprised four exercises carried out biweekly for 8 weeks: a snatch from a squatting position, a bench-press, a half-squat, and a clean and jerk, using a certified weightlifting bar (Table 1). During the first 2 weeks, 3 sets of 6–10 repetitions of each exercise were performed. The initial load corresponded to 55% of the individual’s 1RM, and 3 min of rest were allowed between each set. For the final week, the volume was gradually increased (up to 3–4 sets of 3–4 repetitions) at an intensity of effort from 80% to 85% of 1RM.

### 2.12. Statistical Analyses

Statistical processing was conducted using the “Statistical Package for Social Sciences” (SPSS ver. 25.0, IBM, Armonk, NY, USA). Descriptive statistics were computed for dependent variables. Differences between groups (EG vs. CG) and sessions (pre vs. post) were assessed using a two-factorial (time, group) univariate general linear model. Effect sizes (partial eta squared, η_p_^2^; Richardson, 2011) were calculated for the ANOVA main effects. Differences between means (group, time and group-time effects) were interpreted based on the recommendations of Cohen [23]: *p* < 0.05, η_p_^2^ > 0.14, d ≥ 0.5, and Δd ≥ 1.0. The effect size was calculated as the mean difference between examinations divided by the pooled standard deviation for each variable [24].

## 3. Results

The majority of parameters (7) displayed an excellent relative reliability (ICC ≥ 0.75). Six of seven parameters (92%) also showed an excellent absolute reliability. Only the snatches had a weaker relative reliability (ICC = 0.83; 0.41–0.94) (Table 2).

The two participant groups did not differ significantly in terms of their initial physical characteristics (EG—age: 21.0 ± 0.7 years, body mass: 86.5 ± 5.0 kg, body height: 1.82 ± 0.09 m, body fat percentage: 13.8 ± 0.6%; CG—age: 20.6 ± 0.5 years, body mass: 80.0 ± 8.5 kg, body height: 1.84 ± 0.07 m, body fat percentage: 12.7 ± 1.7%) (Table 3). Playing positions were as follows: EG—goalkeepers, *n* = 2; pivots, *n* = 3; backs, *n* = 4; wings, *n* = 2; CG—goalkeepers, *n* = 2; pivots, *n* = 2; backs, *n* = 3; and wings, *n* = 4. Six subjects were left-handed.

### 3.1. Changes in the Control Group and Experimental Group

#### 3.1.1. Changes in the CG during the 8-Week Trial

Members of the control group did not show any relevant positive effects over the 8-week trial (Table 3 and Table 4).

Clean and jerk scores (Figure 2) showed a small deterioration (d = −0.53), and improvements in snatch (d = 0.17, Figure 3) and upper limb muscle volume (d = 0.23, Figure 4) were very small, with effect sizes ranging from −0.57 (clean and jerk) to 0.23 (limb muscle).

#### 3.1.2. Changes in the EG during the 8-Week Trial

The experimental group showed substantial intervention effects for all measures of performance (Table 3), with effect sizes ranging from 1.40 (throwing velocity standing shot, around a 23% gain) to d = 4.84) (medicine ball throw, a gain of more than 40%). The upper limb muscle volume increased by ~22% (d = 7.00; Table 2, Figure 2) and body mass was the only parameter in the experimental group with a nonrelevant change (d = 0.44).

### 3.2. Comparison of Experimental vs. Control Group

Significant interaction effects were detected for all anthropometric parameters (Table 2). Interaction effects for strength and performance measures ranged from η_p_^2^ = 0.595 (pull-over) to η_p_^2^ = 0.887 (medicine ball throw). 1RM maximum strength and throwing velocity showed consistent time effects (η_p_^2^range: 0.530–0.870) and group × time effects (η_p_^2^range: 0.595–0.887). We found relevant intergroup differences (Δd ≥ 1.0) for all parameters, especially for the performance parameter clean and jerk (Figure 2), jump shot (Figure 5), and running shot (Figure 5). The largest intergroup difference (Δd = 6.65) was for the medicine ball throw (d_EG_ = 6.44 vs. d_CG_ = −0.21), and the smallest (Δd = 1.61) for the standing shot (d_EG_ = 1.40 vs. d_CG_ = −0.21). The largest intergroup difference of anthropometric parameters (η_p_^2^ = 0.920; Δd = 8.59) was for upper limb muscle volume (Figure 6).

## 4. Discussion

The current investigation assessed the impact of an 8-week weightlifting training program on maximal strength of the upper limbs, ball-throwing velocity, and muscle volume in handball players, relative to their standard training protocol. Our findings substantiate the hypothesis that in-season weightlifting training can substantially enhance the maximal strength of the upper limb, ball-throwing velocity, and arm muscle volumes.

Previous studies have investigated the effects of dynamic resistance exercises [1,2,7,8,25], circuit training [26], plyometric training [27], and elastic band training [9], but this is the first investigation to examine the improvement in explosive muscular performance of upper limbs utilizing weightlifting loads in exercises such as the power clean and jerk and power snatches.

Longer contraction durations allowed heavier loads, a tactic suited to maximizing and enhancing strength [10,28], in part by reducing the sensitivity of the Golgi-tendon organs and improving the synchronization of motor unit firing [28].

### 4.1. Muscle Volume

Neural adaptations were not measured, but the substantial increases in muscle volume suggest that hypertrophy of the arm muscles contributed substantially to the gains in performance [9,17], even though the total body mass decreased more in the EG than in the CG. The decrease in fat percentage was greater for the EG, because their total workload was greater than that of the CG (Table 3). The gains in muscle volume of the upper limbs (22%) were higher than those seen during heavy resistance training in elite junior handball players 19% [10], although Hermassi et al. [29] also found gains of 27% in upper limb muscle volume of handball players over 12 weeks of specific resistance training (throwing movements with a medicine ball).

The development of large forces is essential to the complex adaptive processes of muscle tissue remodeling and reshaping, and muscle growth [30,31]. It stimulates cell receptors, membranes, and muscle growth factors, resulting in an increase in protein turnover (synthesis and degradation) and the accretion of muscle protein [10,28].

### 4.2. Maximal Strength

Competitive performance in handball depends on the ability to exert force at the speeds required in sprinting, jumping, and throwing [7,8,25,29]. The prolonged contractions required by the present training regimen seem well-suited to improving muscle strength and power. Others [1,7,25] have noted that resistance training increased upper extremity strength by 23% and leg extensor strength by 12%. The gains of 14% and 23% for the bench-press and pull over, respectively, were greater than those observed by Gorostiaga et al. [1] and Hermassi et al. [8]. The increases of strength of 19% and 22% for the snatch and clean and jerk measures match the observations of Tricoli et al. [11] in male physical education students, and also concur with other studies of plyometric training [26,27], but are less than the gains of 24% and 20% seen by Hermassi et al. [4] following 12 weeks of biweekly in-season weightlifting training of male handball players. Other scholars have reported significant increases in maximal strength of the upper limbs [32] and the lower limbs after 3–16 weeks of strength training [33], and Johns et al. [17] described a 3% increase in upper limb muscle strength in a sample of swimmers following resistance training.

### 4.3. Ball-Throwing Velocity

Elite handball players achieve higher throwing velocities than their lower level peers [2,7], and it is thus important to playing ability that velocities in all three types of throw improved over the present 8-week intervention (Figure 4, Figure 5 and Table 3). Hermassi et al. [7] also noted an increase in all three types of throw following 8 weeks of heavy resistance training, improvements also followed an 8-week biweekly course of plyometric training [27] and eight weeks of triweekly upper limb resistance programs [34]. It is easy to increase workload by gradually increasing weight throwing [30]. The present study shows this clearly. Due to the high load, a small number of repetitions and a limited training time seem to have been sufficient to achieve an optimal response. Our findings are in line with Gorostiaga et al. [25], who described a statistically significant improvement of handball throwing velocity after 6 weeks of heavy upper limb resistance training. However, the training exercises utilized by Goriostoga et al. [25] (the supine bench press, half-squat, knee flexion curl, leg-press, and pec-dec) were quite different from those adopted in the present study.

The EG enhanced their ball speed in standing, running, and jump throws, with gains of 18% and 20% in jump shot and 3-step running throws, respectively. Such gains could be anticipated from earlier studies with lesser workloads and with regular weighted balls [7,26].

### 4.4. Practical Applications

The present investigation indicates that 8 weeks of biweekly in-season training using bench press, pull-over, snatch, and clean-and-jerk weightlifting had a substantial impact on several parameters important to handball performance, including throwing ball velocity, maximal strength, and the muscle volume of the upper limbs. Furthermore, the data suggest that such an intervention could be implemented within the traditional training regimen without adversely affecting other aspects of performance.

Thus, handball coaches should be encouraged to incorporate in-season weightlifting into the training schedule of their teams, as a simple and practical method of enhancing their playing ability. There remains scope for studies that explore the relative contributions of muscle hypertrophy and other potential neuromuscular mechanisms to the observed enhancement of performance.

## 5. Conclusions

Weightlifting exercise offer a stimulus that should be included in any resistance-training program for healthy handball players, who require quick, powerful movements. However, biweekly in-season weight training improved the muscle volume, maximal strength of the upper limbs, and ball throwing velocity in healthy handball players relative to their standard training program. We would also encourage further investigation of the many potential neuromuscular mechanisms contributing to the observed enhancements of anthropometric and physical performance.

## Figures and Tables

**Figure 1 ijerph-16-04520-f001:**
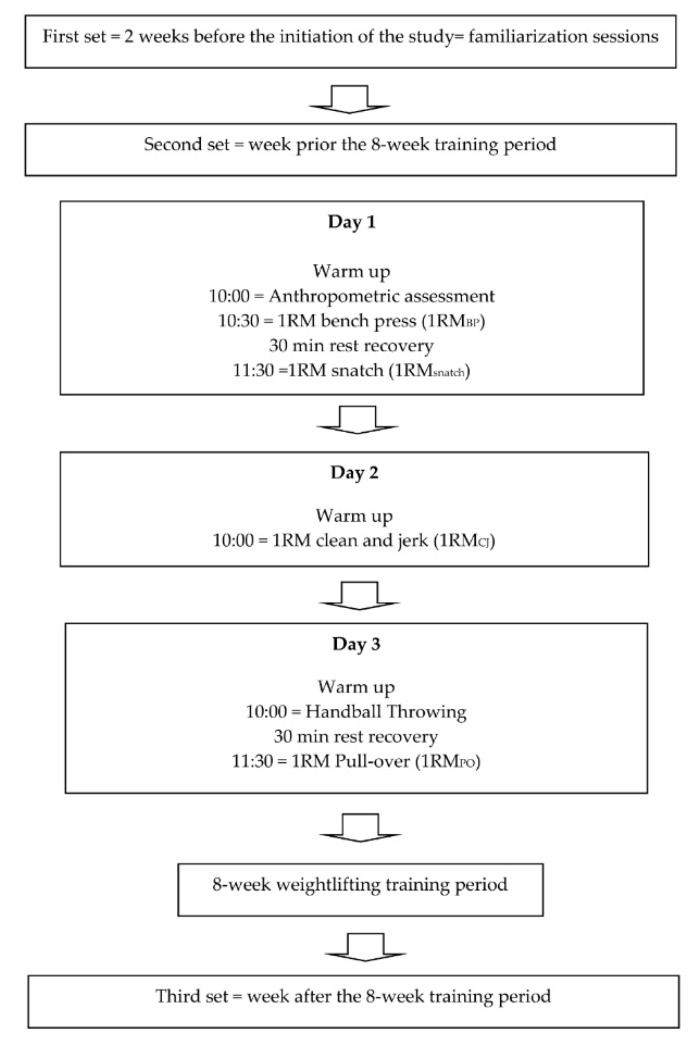
Diagram showing testing schedule.

**Figure 2 ijerph-16-04520-f002:**
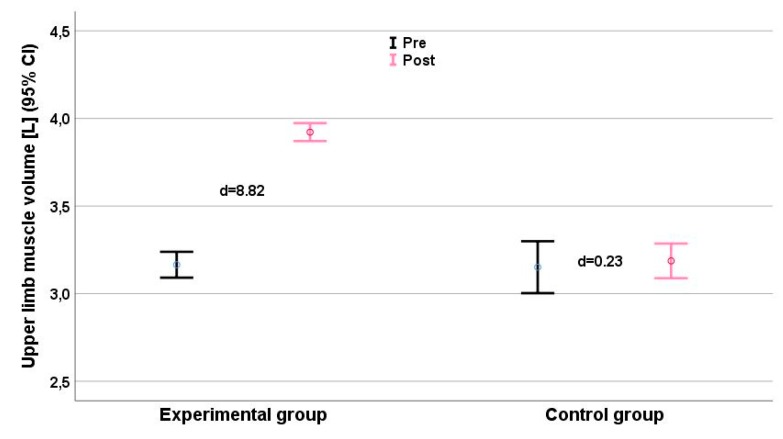
Clean-and-jerk performance (kg) before (black) and after (grey) the intervention. Effect size d for each group (plus means improvement, minus means deterioration) is given. CI = Confidence Interval.

**Figure 3 ijerph-16-04520-f003:**
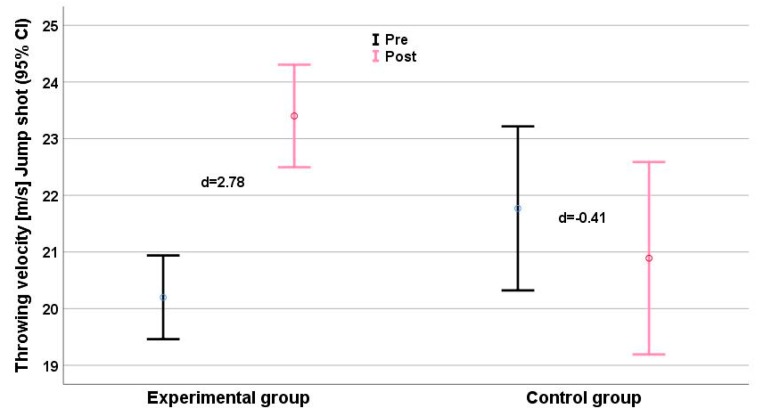
Snatched performance (kg) before (black) and after (grey) the intervention. Effect size d for each group (plus means improvement, minus means deterioration) is given. CI = Confidence Interval.

**Figure 4 ijerph-16-04520-f004:**
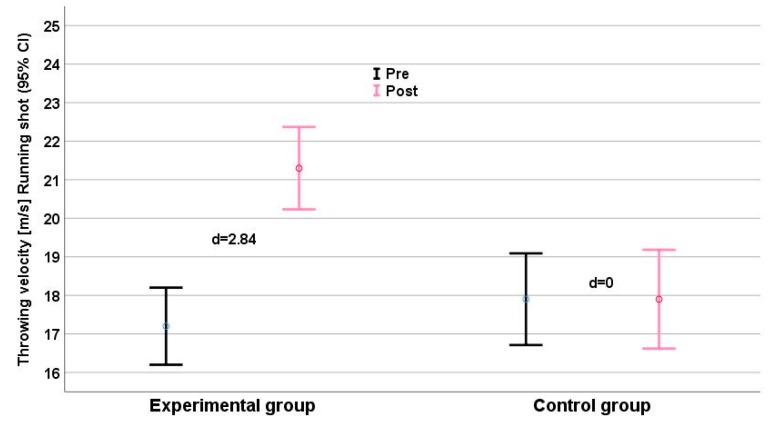
Upper limb muscle volume (L) before (black) and after (grey) the intervention. Effect size d for each group (plus means improvement, minus means deterioration) is given. CI = Confidence Interval.

**Figure 5 ijerph-16-04520-f005:**
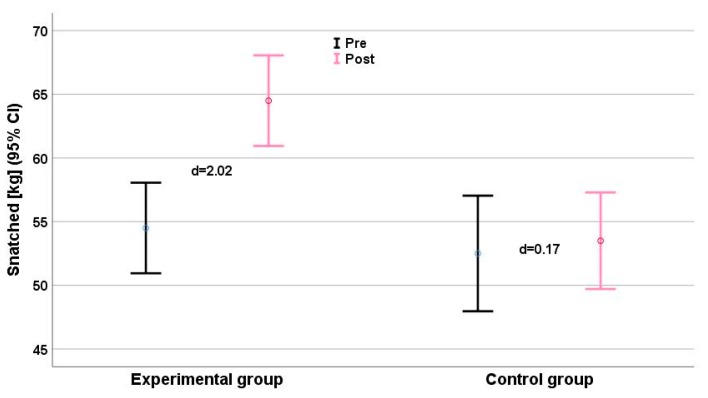
Jump shot performance (m/s) before (black) and after (grey) the intervention. Effect size d for each group (plus means improvement, minus means deterioration) is given. CI = Confidence Interval.

**Figure 6 ijerph-16-04520-f006:**
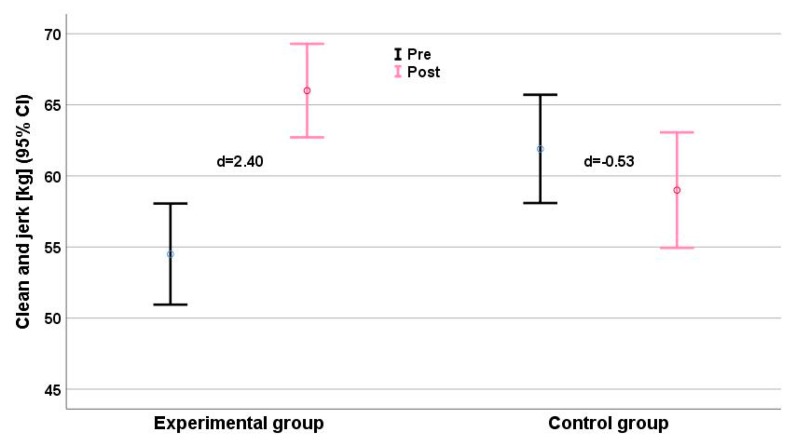
Running shot performance (m/s) before (black) and after (grey) the intervention. Effect size d for each group (plus means improvement, minus means deterioration) is given. CI = Confidence Interval.

**Table 1 ijerph-16-04520-t001:** Details of 8-week weightlifting training program. %1RM and repetitions × sets are indicated.

**Exercises**	**Session 1**	**Session 2**	**Session 3**	**Session 4**	**Session 5**	**Session 6**
Clean and Jerk	55: 3 × 6	55: 3 × 8	60: 3 × 6	60: 3 × 8	60: 4 × 6	60: 4 × 6
Bench press	55: 3 × 6	55: 3 × 8	60: 3 × 6	60: 3 × 8	60: 4 × 6	60: 4 × 6
Snatches	55: 3 × 8	55: 3 × 10	60: 3 × 8	60: 3 × 10	60: 4 × 8	60: 4 × 6
Pull-over	55: 3 × 8	55: 3 × 10	60: 3 × 8	60: 3 × 10	60: 4 × 8	60: 3 × 6
**Exercises**	**Session 7**	**Session 8**	**Session 9**	**Session 10**	**Session 11**	**Session 12**
Clean and Jerk	65: 3 × 6	65: 4 × 6	70: 3 × 5	70: 4 × 5	75: 3 × 5	70: 3 × 6
Bench press	65: 3 × 6	65: 4 × 6	70: 3 × 5	70: 4 × 5	75: 3 × 5	60: 4 × 6
Snatches	65: 3 × 8	65: 4 × 8	70: 3 × 5	70: 4 × 6	75: 3 × 6	60: 4 × 6
Pull-over	65: 3 × 8	65: 4 × 8	70: 3 × 6	70: 4 × 6	75: 3 × 6	70: 2 × 6
**Exercises**	**Session 13**	**Session 14**	**Session 15**	**Session 16**		
Clean and Jerk	75: 4 × 4	80: 3 × 3	80: 3 × 4	85: 3 × 3		
Bench press	75: 4 × 4	80: 3 × 3	80: 3 × 4	85: 3 × 3		
Snatches	75: 4 × 5	80: 3 × 3	80: 4 × 3	85: 4 × 3		
Pull-over	75: 3 × 5	80: 3 × 3	80: 4 × 3	85: 4 × 3		

**Table 2 ijerph-16-04520-t002:** Initial two-week reliability of data for handball players (*n* = 20).

Tests	Session One Mean ± SD	Session Two Mean ± SD	ICC (95% CI)
standing shot (m/s)	23.9 ± 2.86	23.4 ± 2.78	0.80 (0.50–0.92)
jump shot (m/s)	21.0 ± 1.76	19.7 ± 1.93	0.82 (0–0.95)
running shot (m/s)	17.6 ± 1.54	16.9 ± 1.55	0.90 (0.50–0.97)
snatches (kg)	53.5 ± 5.64	50.5 ± 6.67	0.83 (0.41–0.94)
clean and jerk (kg)	58.2 ± 6.29	52.0 ± 6.37	0.58 (0–0.85)
pull over (kg)	37.9 ± 4.84	36.8 ± 4.94	0.89 (0.73–0.96)
bench press (kg)	73.3 ± 5.20	69.3 ± 4.94	0.74 (0–0.92)

**Table 3 ijerph-16-04520-t003:** Anthropometric characteristics of study participants (mean ± SD) before and after 8-week trial. Significant interaction effects (η_p_^2^ ≥ 0.14) and effect sizes (d ≥ 0.50) are highlighted in bold.

Parameter	Experimental Group (*n* = 10)			Control Group (*n* = 10)			Variance Analysis/Effects			
	Pre	Post		Pre	Post		Time		Group × Time	
			d			d	*p*	η_p_^2^	*p*	η_p_^2^
Body mass (kg)	86.5 ± 5.0	84.4 ± 4.6	0.44	80.0 ± 8.5	80.0 ± 8.1	0	0.002	0.433	**0.002**	**0.433**
Body fat (%)	13.8 ± 0.6	13.4 ± 0.6	**0.66**	12.7 ± 1.7	12.7 ± 1.6	0	0.001	0.462	**0.003**	**0.391**
Upper limb muscle volume (L)	3.2 ± 0.1	3.9 ± 0.1	**7.00**	3.1 ± 0.2	3.2 ± 0.1	0.23	<0.001	0.933	**<0.001**	**0.920**

**Table 4 ijerph-16-04520-t004:** Velocity and maximal strength performance (mean ± SD) before and after 8-week trial. Significant interaction effects (η_p_^2^ ≥ 0.14) and effect sizes (d ≥ 0.50) are highlighted in bold.

Parameter	Experimental Group (*n* = 10)			Control Group (*n* = 10)			Variance Analysis/Effects			
	Pre	Post		Pre	Post		Time		Group × Time	
			d			d	*p*	η_p_^2^	*p*	η_p_^2^
	**Throwing velocity**									
Standing shot (m/s)	23.7 ± 3.6	29.0 ± 4.0	**1.40**	24.1 ± 2.1	23.7 ± 1.8	−0.21	<0.001	0.530	**<0.001**	**0.603**
Jump shot (m/s)	20.2 ± 1.0	23.4 ± 1.3	**2.78**	21.8 ± 2.0	20.9 ± 2.4	−0.41	<0.001	0.530	**<0.001**	**0.777**
Running shot (m/s)	17.2 ± 1.4	21.3 ± 1.5	**2.84**	17.9 ± 1.7	17.9 ± 1.8	0	<0.001	0.771	**<0.001**	**0.771**
Medicine ball throw (m)	18.5 ± 1.9	26.0 ± 1.2	**4.84**	18.1 ± 1.6	17.8 ± 1.2	−0.21	<0.001	0.870	**<0.001**	**0.887**
	**Maximal strength performance (kg)**									
Snatch	54.5 ± 4.9	64.5 ± 5.0	**2.02**	52.5 ± 6.3	53.5 ± 5.3	0.17	<0.001	0.761	**<0.001**	**0.681**
Clean and jerk	54.5 ± 5.0	66.0 ± 4.6	**2.40**	61.9 ± 5.3	59.0 ± 5.7	−0.53	<0.001	0.537	**<0.001**	**0.764**
Pull-over	39.3 ± 4.8	48.0 ± 5.2	**1.75**	36.5 ± 4.7	36.5 ± 2.4	0	<0.001	0.595	**<0.001**	**0.595**
Bench press	73.5 ± 4.7	83.5 ± 4.7	**2.11**	73.0 ± 5.9	72.5 ± 5.4	−0.09	<0.001	0.862	**0.001**	**0.884**

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
