# Peer review of "In-Season Weightlifting Training Exercise in Healthy Male Handball Players: Effects on Body Composition, Muscle Volume, Maximal Strength, and Ball-Throwing Velocity"

_ijerph, 2019, doi:10.3390/ijerph16224520_

Round 1

Reviewer 1 Report

The author's inquired if adding a biweekly weightlifting session over an 8-week program would improve performance compared to traditional training over the 8 weeks. Accordingly, twenty subjects were randomly assigned to one of two groups (experimental, control).  The data demonstrate that the hypothesis was confirmed, and that weightlifting increased many performance areas.  The strength of this manuscript is the intervention. It was well-designed, and carefully controlled. Overall, I think that this manuscript has some strengths, and some weaknesses.  However, I think the weaknesses are easy to address, and I have outlined them below.

-Page 2, lines 64-76. Where there any inclusion/exclusion criteria? A healthy check-up was inclusionary, as is highlighted in the text.  Anything else?  Supplements?  Medications? Diabetes? Cardiovascular health?

-Page 3, line 96. 1RM hasn't been defined as of yet.

-Page 3, lines 114 and 119. The authors use 'participant', and then 'subject' and then back to 'participant'.  Please, for the sake of the reader, be consistent.

-Page 3, line 120.  '1RMBP and 1RMsnatch' hasn't been defined yet.

-Page 3/4. Can the authors discuss the 1RM procedure a bit more? What was the rest break between attempts? How many attempts?

-Page 4, line 158.  '2-3 at 100%' how is that possible?  Wouldn't 100% of the the 1RM be 1? 

-Page 6, line 225. SPSS is no longer in Chicago, please update.      

Author Response

Review Report (Reviewer 1)

Comments and Suggestions for Authors

Reviewer

The author's inquired if adding a biweekly weightlifting session over an 8-week program would improve performance compared to traditional training over the 8 weeks. Accordingly, twenty subjects were randomly assigned to one of two groups (experimental, control).  The data demonstrate that the hypothesis was confirmed, and that weightlifting increased many performance areas.  The strength of this manuscript is the intervention. It was well-designed, and carefully controlled. Overall, I think that this manuscript has some strengths, and some weaknesses.  However, I think the weaknesses are easy to address, and I have outlined them below.

Author Response (AR)

Dear Reviewer 1

Thank you for your careful perusal of our manuscript and your helpful comments. We have addressed your comments in our point-by-point statements and made changes to the manuscript whenever necessary. We hope that our revision improved the quality of our manuscript so that it is now suitable for publication in the frontier journal.

Responses to reviewer-1  are highlighted in yellow,

Responses to reviewer-2 are highlighted in green,

Responses to reviewer-3 are highlighted in blue,

Common responses to all reviewers are underlined

Reviewer

Page 2, lines 64-76. Where there any inclusion/exclusion criteria? A healthy check-up was inclusionary, as is highlighted in the text.  Anything else?  Supplements?  Medications? Diabetes? Cardiovascular health?

Author Response (AR)

Thank you for your suggestion. The following sentences was added in the text file (Procedures and evaluations):

Line 79-80: “We excluded individuals undergoing any type of medical treatment and those who were obese (body mass index > 30 kg/m2)’.

Reviewer

Page 3, line 96. 1RM hasn't been defined as of yet.

Author Response (AR)

You can find the definition in line 29-30 (1RM (repetition maximum)) and Line 102-103:’’ … at 60–85% of the individual's personal one repetition maximum [1RM], intended’’

Reviewer

Page 3, lines 114 and 119. The authors use 'participant', and then 'subject' and then back to 'participant'.  Please, for the sake of the reader, be consistent.

Author Response (AR)

We replaced subject by participant.

Reviewer

-Page 3, line 120.  '1RMBP and 1RMsnatch' hasn't been defined yet.

Author Response (AR)

Thank you for your suggestion. The following paragraph was added in the text file:

Line 123-126: On the first definitive test day, anthropometric measurements were completed, followed by the 1RM bench press (1RMBP) and 1RM snatch (1RMsnatch) tests. On the second day, 1RM clean and jerk (1RMcj) tests were assessed. On the third day, handball throwing and the 1RM pull-over (1RMPO) were evaluated (Figure 1).

Reviewer

Page 3/4. Can the authors discuss the 1RM procedure a bit more? What was the rest break between attempts? How many attempts?

Author Response (AR)

Thank you for your relevant suggestion. We added written 1RM procedure in full.

Line 159-164

Line 170-175

Line 185-190

Line 194-205

Reviewer

Page 4, line 158.  '2-3 at 100%' how is that possible?  Wouldn't 100% of the the 1RM be 1?

Author Response (AR)

Thank you for your relevant remarks. The following paragraph was corrected in the text file:

Line 169-170: ‘’Definitive lifts were performed as follows: 2 at 70%, 2 at 80% 1 at 90%, 1 at 95%, and 2-3 at 100% of the one repetition maximum (1RM) [14]’’.

Reviewer

Page 6, line 225. SPSS is no longer in Chicago, please update.

Author Response (AR)

Line 244-245: Statistical processing was conducted using the “Statistical Package for Social Sciences” (SPSS ver. 25.0, IBM, Armonk, NY, USA).

Reviewer 2 Report

Thank you to the authors for submitting their manuscript to Environmental Research and Public Health; I enjoyed reading it.

There some suggestions that I think will provide clarity to the reader (outlined below).

Good luck with your amendments and I look forward to seeing the revised version.

SPECIFIC COMMENTS

Title manuscript:

Please add “male” in the title manuscript to clarify the gender of population.

Abstract:

Revise % of 1-RM of dynamic weighlifting exercise. I have seen that exercises were at 55-90% of 1-RM (table 1).

Material and Methods:

I wasn’t sure exactly what each exercise was. Perhaps a picture or description would help, even as an additional supplementary file. I (and other readers) might wish to implement this intervention with male handball players in the future and more detail would help here

Please ensure you put “one repetition maximum” after you spell out 1RM for the first time. (line 95)

Please add a gap after “velocity”. (line 97)

What kind of shoes? (line 100-103)

Add apart from performing a general warm-up, a specific warm-up is also performed in each test. (line 109-112)

It is very interesting to assess test-retest reliability. But, where are these results? (line 115)

Please ensure you put “maximum one-repetition eccentric-concentric bench press action” after you spell out 1RMBP for the first time. (line 120)

Indicate how this sample was randomized. (line 126)

Add weight ball was 3kg. Why 3kg? (line 186)

What mean “55” in table 1. Please more clear (i.e. 55% 1RM)

Please replace “control group” with “CG”. (line 235)

Capital letter after “3.” (line 240).

Please revise if it is correct medicine ball throw of “d” result in experimental group. It is appears me so elevate, maybe could be “3.94”.

Please separate “0” and -0.53. (line 244).

Please replace “experimental group” with “EG”. (line 261)

I do not understand why do you indicate “figure 1, 4 and 5”. In this section you comment inter-group results and these figures show intra-group analyse. Please indicate only table 2 and 3 where are showed inter-group results. (line 266).

Discussion

Hypertrophy o? (line 299)

Author Response

Review Report (Reviewer 2)

Comments and Suggestions for Authors

Reviewer

Thank you to the authors for submitting their manuscript to Environmental Research and Public Health; I enjoyed reading it. There some suggestions that I think will provide clarity to the reader (outlined below).

Good luck with your amendments and I look forward to seeing the revised version.

Author Response (AR)

Thank you for this positive review. We found the criticisms and recommendations positive and very constructive. We much appreciate the work that you have put into reviewing this manuscript, and we think that the quality of the manuscript has been substantially improved as a result of your comments. The changes made to the paper are highlighted in the new version. As you will see, we have carefully considered all your suggestions in order to improve the manuscript.

Responses to reviewer-1  are highlighted in yellow,

Responses to reviewer-2 are highlighted in green,

Responses to reviewer-3 are highlighted in blue,

Common responses to all reviewers are underline.

Reviewer

SPECIFIC COMMENTS

Title manuscript:

Please add “male” in the title manuscript to clarify the gender of population.

Author Response (AR)

Line 2:

Thank you for your relevant remark. We added ‘’male’’ in the title as suggested

Line 3: …… male handball players..’’

Reviewer

Abstract:

Revise % of 1RM of dynamic weighlifting exercise. I have seen that exercises were at 55-90% of 1RM (table 1).

Author Response (AR)

Thank you for your relevant remark. We have replaced in the line 29 dynamic by ‘’explosive’’ weightlifting exercise.

Reviewer

Material and Methods:

I wasn’t sure exactly what each exercise was. Perhaps a picture or description would help, even as an additional supplementary file. I (and other readers) might wish to implement this intervention with male handball players in the future and more detail would help here.

Author Response (AR)

According to the reviewer’s suggestion, we have created and added a new figure 1 description of all exercise steps.

Reviewer

Please ensure you put “one repetition maximum” after you spell out 1RM for the first time. (line 95)

Author Response (AR)

Thank you for this hint. We check everything in this direction.

Reviewer

Please add a gap after “velocity”. (line 97)

Author Response (AR)

Line 97-100:

We added a gap after ‘’velocity’’ as suggested in the line 104

Reviewer

What kind of shoes? (line 100-103)

Author Response (AR)

Thank you for your relevant suggestion. We have added the following sentence in the text:

Line 100-101: ‘’They wore handball shoes to adapt and fit their specific efforts to handball activity’’.

Reviewer

Add apart from performing a general warm-up, a specific warm-up is also performed in each test. (line 109-112)

Author Response (AR)

We added this sentence in line 113.

Reviewer

It is very interesting to assess test-retest reliability. But, where are these results? (line 115)

Author Response (AR)

Thank you for your relevant suggestion. We added written ICC in full in the new table 2

Reviewer

Please ensure you put “maximum one-repetition eccentric-concentric bench press action” after you spell out 1RMBP for the first time. (line 120)

Author Response (AR)

We ensure this in order to be consistent in the line 123-126.

Reviewer

Indicate how this sample was randomized. (line 126)

Author Response (AR)

Thank you for your suggestion. We randomized the sample using a random number generator.

Line 132-133: ‘’…A sample of 20 male handball players was randomly assigned between..’’

Reviewer

Add weight ball was 3kg. Why 3kg? (line 186)

Author Response (AR)

Thank you for your remark this is a mistake and we corrected as suggested.

Reviewer

What mean “55” in table 1. Please more clear (i.e. 55% 1RM)

Author Response (AR)

We added an explanation in the legend of the table in the 241.

Reviewer

Please replace “control group” with “CG”. (line 235)

Author Response (AR)

We replaced “control group” with “CG”

Line 263: ‘’3.1.1. Changes in the CG during the 8-week trial’’

Reviewer

Capital letter after “3.” (line 240).

Author Response (AR)

I don’t understand what the reviewer means.

I cannot find “3” in line 240: 

Line 269-270:Table 3. Anthropometrics characteristics of study participants (mean ± SD) before and after 8-week trial. Significant interaction effects (hp2≥0.14) and effect sizes

Reviewer

Please revise if it is correct medicine ball throw of “d” result in experimental group. It is appears me so elevate, maybe could be “3.94”.

Author Response (AR)

Thanks a lot for this hint. We recalculated and corrected the effect size (6.44 ® 4.84 and 8.82 ® 7.00).

Reviewer

Please separate “0” and -0.53. (line 244).

Author Response (AR)

Thank for your suggestion but we don’t understand what the reviewer means in Table 4.

Reviewer

Please replace “experimental group” with “EG”. (line 261)

Author Response (AR)

Replaced as suggested

Line 294: ‘’3.1.2. Changes in the EG during the 8-week trial’’

Reviewer

I do not understand why do you indicate “figure 1, 4 and 5”. In this section you comment inter-group results and these figures show intra-group analyse. Please indicate only table 2 and 3 where are showed inter-group results. (line 266).

Author Response (AR)

The content of both tables is identical. It was only a problem of the layout. We corrected the layout of table 2 based on table 3 (column width of variance analysis/ effects) in order to improve the readability.

Reviewer

Discussion

Hypertrophy o? (line 299).

Author Response (AR)

Thank you, corrected “hypertrophy of” in the line 385.

Reviewer 3 Report

Introduction

This section in too short. Please present and explore more the literature the existing literature (as you introduced in the first paragraph) in this strength and conditioning in handball, to lead the reader to understand the real lack in this research area.

Material and Methods

In general, this session is quite confusing. Please consider to reorganize all the subtitles.

Line 67-68 – the authors stated that “parents/legal guardians were thoroughly advised about the experimental design of the present study and its potential risks and then signed their written informed consent” however, all the participants had more than 20 years. It was this really necessary? Line 69-76 – The indication that you used a questionnaire, the examinations conducted, and the participants characteristics were not in accordance to the subtitle of this session. Please consider removing and add or adjust to another session. Line 78-81 – That information seems to make part of the results, not methods. Line 88 – why including gymnastics? Line 86-89 – please clarify how did you perform that planification, it was all in the same week? Don’t you think that all those methodologies could be concurrent trainings? Line 90-91 – “included strength and conditioning routines (the remaining 40% of session time)”, thus, the CG also performed strength and conditioning trainings? Line 93-94 – The biweekly strength training sessions were included in the 3 training per week indicated in line 86? And the CG performed also those sessions 3 times per week, but the EG had 2 more? Please clarify. Line 102-103 – The test procedures were conducted when the athletes are no longer in training sessions (untrained?)? Line 109-110 – please use always the same form of writing: 30-m or 30 m. Line 109-113 – could it have been too much for a warmup? What was the average duration? Line 122-130 – I think that this information is duplicated and did not discuss the real experimental approach to the problem. Line 131-135 – your real aim of the study was to observe if this intervention increase the muscle volume or if it increase strength? Line 138-144 – please clarify why you use the bench press to analyse strength and the warm-up was exclusively with the lower limbs? Line 158-160 – after the test, athletes performed a warm-up? How many minutes they rested between sets? Line 209 – I am very concerned about the type of training chosen, as 3-10 repetitions seems an interval too big for me, I am wondering if the goal was strength gain or muscle hypertrophy. Line 214 – 30 seconds of rest to increase strength?

Results

Your test to understand the effects of this intervention was muscle volume because you want to increase muscle hypertrophy? Or your focus was strength gains and, therefore, you want to improve velocity?

Discussion

Line 291-293 – In fact, in my searches I found at least one study:

Hermassi, S., Schwesig, R., Aloui, G., Shephard, R. J., & Chelly, M. S. (2019). Effects of Short-Term In-Season Weightlifting Training on the Muscle Strength, Peak Power, Sprint Performance, and Ball-Throwing Velocity of Male Handball Players. The Journal of Strength & Conditioning Research.

- Line 298-300 – here the authors indicating that gain in muscle volume is the same as strength gains, and this is not true.

Author Response

Review Report (Reviewer 3)

Comments and Suggestions for Authors

Reviewer

Introduction

This section in too short. Please present and explore more the literature the existing literature (as you introduced in the first paragraph) in this strength and conditioning in handball, to lead the reader to understand the real lack in this research area.

Author Response (AR)

Thank you for your careful perusal of our manuscript and your helpful comments. We have addressed your comments in our point-by-point statements and made changes to the manuscript whenever necessary. We hope that our revision improved the quality of our manuscript so that it is now suitable for publication in the frontier journal.

Responses to reviewer-1  are highlighted in yellow,

Responses to reviewer-2 are highlighted in green,

Responses to reviewer-3 are highlighted in blue,

Common responses to all reviewers are underlined

Concerning your relevant suggestion. We have added the following paragraph in the introduction:

Line 46-50: ‘’Both strength and power can be enhanced by resistance training [3, 4]. Some studies have found associated improvements in explosive actions [2,3,4], but others have seen no significant increases [4], suggesting that strength training may lack elements important to explosive movements, including eccentric overloading, segmental coordination, and specificity of joint angles and angular velocities [3]’’.

Reviewer

Material and Methods

In general, this session is quite confusing. Please consider to reorganize all the subtitles.

Author Response (AR)

According to the reviewer’s suggestion we have reorganize the material and methods section and created new figure description of all exercise and protocols steps.

Reviewer

Line 67-68 – the authors stated that “parents/legal guardians were thoroughly advised about the experimental design of the present study and its potential risks and then signed their written informed consent” however, all the participants had more than 20 years. It was this really necessary?

Author Response (AR)

We corrected this statement as suggested:

Line 73-75: ‘’Participants were thoroughly advised about the experimental design of the present study and its potential risks and benefits, and they then signed their written informed consent’’.

Reviewer

Line 69-76 – The indication that you used a questionnaire, the examinations conducted, and the participants characteristics were not in accordance to the subtitle of this session. Please consider removing and add or adjust to another session.

Author Response (AR)

Thank you for this valuable advice. We removed this paragraph at the start of the following session adjusted to another session:

Procedures and evaluations

Line 78- 88: ‘’A questionnaire regarding socio-demographic and anthropometric parameters (such as medical history, age, height, body mass, training characteristics, injuries, playing position, performance experience and level) was first administered. The team physician also carried out an initial examination, focusing on orthopedic conditions and clinical disorders that might preclude participation in resistance exercises. Twenty healthy male handball players from the Second National League (age: 21.2 ± 0.7 years, height: 1.83 ± 0.08 m, body mass: 83.3 ± 7.5 kg, body fat: 13.2 ± 1.4%, mean playing experience 10.1 ± 0.5 years), were randomly allocated between an experimental (EG) and a control (CG) group’’.

Reviewer

Line 78-81 – That information seems to make part of the results, not methods. Line 88 – why including gymnastics?

Author Response (AR)

We removed this paragraph to the results:

 Line 256-261: ‘’The two groups did not differ significantly in terms of initial physical characteristics (EG: age: 21.0 ± 0.7 years, body mass: 86.5 ± 5.0 kg, body height: 1.82 ± 0.09 m, body fat percentage: 13.8 ± 0.6%; CG: age: 20.6 ± 0.5 years, body mass: 80.0 ± 8.5 kg, body height: 1.84 ± 0.07 m, body fat percentage: 12.7 ± 1.7%). Playing positions were as follows: EG: goalkeepers, n=2; pivots, n=3; backs, n=4; wings, n=2; CG: goalkeepers, n=2; pivots, n=2; backs, n=3; and wings, n=4. Six subjects were left-handed’’.

In addition, we deleted “gymnastics”.

Reviewer

Line 86-89 – please clarify how did you perform that planification, it was all in the same week? Don’t you think that all those methodologies could be concurrent trainings?

Author Response (AR)

According to the reviewer’s suggestion we have clarify how we did perform the planification as following:

’’During the 8-week intervention, the CG maintained this standard pattern of training, but twice per week the EG replaced their technical-tactical skill development by a dynamic weightlifting program designed to maximize increases in strength. They wore handball shoes to adapt and fit their specific efforts to handball activity. Training focused upon the extensor muscles of the upper limbs (bench-press, pull-over, snatch, and clean-and-jerk exercises at 60–85% of the individual's personal one repetition maximum [1RM]). All of these exercises required a series of successive eccentric–concentric loaded contractions, conducted at a slow velocity; 3–8 repetitions were carried out per set, and 3–6 sets were interspersed with rest intervals lasting 3 to 4 minutes’’.

Reviewer

Line 90-91 – “included strength and conditioning routines (the remaining 40% of session time)”, thus, the CG also performed strength and conditioning trainings?

Author Response (AR)

Thank you for your remarks. We reworked the following paragraph and added more information about the standard training pattern of CG during strength and conditioning routines:

Reviewer

Line 93-94 – The biweekly strength training sessions were included in the 3 training per week indicated in line 86? And the CG performed also those sessions 3 times per week, but the EG had 2 more? Please clarify.

Author Response (AR)

Corrected as suggested in the planification.

Reviewer

Line 102-103 – The test procedures were conducted when the athletes are no longer in training sessions (untrained?)?

Author Response (AR)

Thank you for you relevant remark. Indeed the tests were conducted at least 3 days after a competition. “5-9 days after the last training session” is now removed.

Reviewer

Line 109-110 – please use always the same form of writing: 30-m or 30 m.

Author Response (AR)

We changed the form of writing (30 m) as suggested in the line 114.

Reviewer

Line 109-113 – could it have been too much for a warmup? What was the average duration?

Author Response (AR)

Thank you for this relevant question. The average duration of the warm up equal to 15 min

Reviewer

Line 122-130 – I think that this information is duplicated and did not discuss the real experimental approach to the problem.

Author Response (AR)

Thank you for your remarks. Corrected as suggested as following:

Line 124-125: ‘’On the third day, the handball throwing and 1RM Pull-over (1RMPO) was evaluated’’.

Reviewer

Line 131-135 – your real aim of the study was to observe if this intervention increase the muscle volume or if it increase strength?

Author Response (AR)

The increasing strength is an indicator/proof for the effectiveness of the intervention. In that sense, it is not a contradiction.

Reviewer

Line 138-144 – please clarify why you use the bench press to analyse strength and the warm-up was exclusively with the lower limbs?

Author Response (AR)

Thank you for your remarks. Corrected as suggested as following:

Line 166-167: ‘’After a 10-min standardized warm-up participants began with a number of warm-up lifts at 40-60% of 1 RM’’.

Reviewer

Line 158-160 – after the test, athletes performed a warm-up? How many minutes they rested between sets?

Author Response (AR)

The warm-up was modified as suggested and the minutes rested between sets was included in the text in the line 169.

Reviewer

Line 209 – I am very concerned about the type of training chosen, as 3-10 repetitions seems an interval too big for me, I am wondering if the goal was strength gain or muscle hypertrophy.

Author Response (AR)

Thank you for this relevant question. We have corrected the missed information as following:

Line 227-228: ‘’Each training session consisted of four different exercises and from 3 to 4 sets of 8–10 repetitions [8]’’.

Reviewer

Line 214 – 30 seconds of rest to increase strength?

Author Response (AR)

Thank you for your remark. Corrected as suggested as following:

Line 232-233: ‘’If the required number of repetitions could not be achieved within a set, 3 minutes of rest were allowed before further attempts’’.

Reviewer

Results

Your test to understand the effects of this intervention was muscle volume because you want to increase muscle hypertrophy? Or your focus was strength gains and, therefore, you want to improve velocity?

Author Response (AR)

The major aim of the current strength training is to enhance strength and thereafter performance. It is well know that muscle strength could be enhanced either by muscle hypertrophy or by neuromuscular adaptations. The current result indicates that upper limbs muscle volume was significantly enhanced after the training program. The gain in performance could be then attributed mainly to muscle hypertrophy. This justification is presented in “Muscle volume” paragraph in the “Discussion section”.

Reviewer

Discussion

Line 291-293 – In fact, in my searches I found at least one study:

Hermassi, S., Schwesig, R., Aloui, G., Shephard, R. J., & Chelly, M. S. (2019). Effects of Short-Term In-Season Weightlifting Training on the Muscle Strength, Peak Power, Sprint Performance, and Ball-Throwing Velocity of Male Handball Players. The Journal of Strength & Conditioning Research.

Author Response (AR)

We agree with the reviewer. Indeed Hermassi et al. (2019) is cited in “Maximal strength” paragraph of the discussion section.

Reviewer

- Line 298-300 – here the authors indicating that gain in muscle volume is the same as strength gains, and this is not true.

Author Response (AR)

Corrected as suggested.

Round 2

Reviewer 3 Report

In my opinion, the introduction session should be strengthened with more studies related to strength training in those athletes and its effects.

Line 168 – If the 1RM was well measured, how could the athletes perform 2-3 repetitions at 100% of 1RM? Specially after performing “2 at 70%, 2 at 80% 1 at 90%, 1 at 95%”?

The text “The barbell was loaded to 90% of the pretest 1RM. Two consecutive tests were carried out, and if 2 repetitions were completed, a load of 5 kg was added after a recovery interval of 3 minutes. When the participant had performed 2 successful repetitions of his pretest RM value, further loads of 1 kg were added after the recovery interval. If the second repetition could not be completed with the new loading, the corresponding load was considered as the individual’s 1RM [14]. Three to 6 attempts were needed to reach the 1RM” is repeated 4 times in the methods, it could be elsewhere and indicated that all the tests followed these procedures.

Line 268 – please consider removing the “s” of “anthropometrics”

Author Response

Review Report (Reviewer 3)

Comments and Suggestions for Authors

Reviewer

1. In my opinion, the introduction session should be strengthened with more studies related to strength training in those athletes and its effects.

Dear Reviewer

Thank you for your careful perusal of our manuscript and your helpful comments. We have addressed your comments in our point-by-point statements and made changes to the manuscript whenever necessary. We hope that our revision improved the quality of our manuscript so that it is now suitable for publication in the frontier journal.

Responses to reviewer-1  are highlighted in yellow,

Author Response (AR)

Thank you for your suggestion we have amended and added in the introduction more paragraph with more studies related to strength training in those athletes and its effects.

Line 43-57

Line 73-77

Reviewer

2. Line 168 – If the 1RM was well measured, how could the athletes perform 2-3 repetitions at 100% of 1RM? Specially after performing “2 at 70%, 2 at 80% 1 at 90%, 1 at 95%”?

Author Response (AR)

Thank you for your relevant remarks. The following paragraph was corrected in the text file:

Line 188- 190: ‘’Definitive lifts were performed as follows: 2 at 70%, 2 at 80% 1 at 90%, 1 at 95% of the one repetition maximum (1RM) with 2-5 minutes rested between sets [14]. The barbell was then loaded to 90% of the pretest 1RM’’.

Reviewer

3. The text “The barbell was loaded to 90% of the pretest 1RM. Two consecutive tests were carried out, and if 2 repetitions were completed, a load of 5 kg was added after a recovery interval of 3 minutes. When the participant had performed 2 successful repetitions of his pretest RM value, further loads of 1 kg were added after the recovery interval. If the second repetition could not be completed with the new loading, the corresponding load was considered as the individual’s 1RM [14]. Three to 6 attempts were needed to reach the 1RM” is repeated 4 times in the methods, it could be elsewhere and indicated that all the tests followed these procedures. 

Author Response (AR)

Corrected as suggested 

Line 156-157: “Three to 6 attempts were needed to reach the 1RM for Bench Press, Bench snatch, Clean and Jerk and Pull-over’’.

Reviewer

4. Line 268 – please consider removing the “s” of “anthropometrics”

Author Response (AR)

Removed as suggested.